# Endoscopic Ultrasound (EUS) Guided Elastography

**DOI:** 10.3390/diagnostics13101686

**Published:** 2023-05-10

**Authors:** Julio Iglesias-Garcia, Daniel de la Iglesia-Garcia, Jose Lariño-Noia, Juan Enrique Dominguez-Muñoz

**Affiliations:** Department of Gastroenterology and Hepatology, Health Research Institute of Santiago de Compostela (IDIS), International Center for Education and Development in Gastroenterology (ICEDiG), University Hospital of Santiago de Compostela, 15706 Santiago de Compostela, Spain

**Keywords:** endoscopic ultrasound, advance imaging, strain elastography, shear wave elastography

## Abstract

Endoscopic ultrasound (EUS) is an essential technique for the management of several diseases. Over the years, new technologies have been developed to improve and overcome certain limitations related to EUS-guided tissue acquisition. Among these new methods, EUS-guided elastography, which is a real-time method for the evaluation of tissue stiffness, has arisen as one of the most widely recognized and available. At present, there are available two different systems to perform an elastographic evaluation: strain elastography and shear wave elastography. Strain elastography is based on the knowledge that certain diseases lead to a change in tissue hardness while shear wave elastography monitored shear-wave propagation and measures its velocity. EUS-guided elastography has shown in several studies high accuracy in differentiating benign from malignant lesions from many different locations, mostly in the pancreas and lymph nodes. Therefore, nowadays, there are well-established indications for this technology, mainly for supporting the management of pancreatic diseases (diagnosis of chronic pancreatitis and differential diagnosis of solid pancreatic tumors) and characterization of different diseases. However, there are more data on new potential indications for the near future. In this review, we will present the theoretical bases of this technology and we will discuss the scientific evidence to support its use.

## 1. Introduction

Endoscopic ultrasound (EUS) has induced an important change in the management of several diseases since its inclusion in clinical routine. In fact, over the years, data are demonstrating that it has changed the clinical approach in more than 50% of cases [1,2,3,4,5,6]. However, standard B-mode imaging cannot always provide the perfect diagnosis, thus needing EUS-guided sampling. The diagnostic yield of EUS-guided sampling has shown to be very high, with a sensitivity ranging from 80 to 85%, and specificities close to 100%. This is mostly due to the development of core needles [7,8]. In this setting, histological needles have supposed a great advance, with the ability to provide real core samples. In fact, it is possible, not only to obtain a cyto-histological diagnosis, but also to obtain detail information on the lesion profile bases on different analyses that can be performed with the sample, such immunostaining or molecular evaluation [9,10,11,12]. However, EUS-guided sampling can be challenging with the necessity to access multiple times into the lesion to obtain the diagnosis [13,14,15] and it can also be associated with adverse events [16,17,18]. In this scenario, new technologies such as contrast-enhanced EUS and EUS-guided elastography have arisen, limiting the need for EUS-guided biopsy of the lesions, but also help in guiding the biopsy to regions with the highest suspicion for malignancy.

Elastography is a real-time technique permitting the analysis of tissue stiffness. Changes in tissue stiffness can be associated with various pathologies, including cancer. Elastography was first developed for the evaluation of lesions visible from the body surface [19,20]. Elastographic evaluation can now be conducted in conjunction with traditional EUS from inside the gastrointestinal tract. Its high accuracy in distinguishing benign from malignant lesions from a wide variety of locations, primarily in the pancreas and lymph nodes, has been demonstrated in many studies. We will go over the technical details and clinical usefulness of EUS elastography.

## 2. Technical Aspects and Methodology of Elastography

Nowadays, two different systems to perform an elastographic evaluation are available: strain elastography and shear wave elastography.

### 2.1. Strain Elastography

The theory behind strain elastography is that certain pathologies, including cancer, induce a change in the elasticity modulus of tissues, which in turn affects how hard the tissues are. The well-known fremitus technique in breast ultrasonography, which shows that healthy breast tissue vibrates more than solid malignant lesions, can be seen as the forerunner of elastography [19]. The technology is based on the recognition of small structure deformations brought about by compression within the B-mode image and the degree of deformation is used as an indicator of the stiffness of the tissue [21].

By registering differences in distortion of the EUS image following the application of slight pressure by the EUS probe, this noninvasive method measures elasticity in real-time, together with physiologic vascular pulsations and respiratory movements which provide the vibrations and compressions necessary for the recording. As stated before, certain pathologic processes, such as inflammation, fibrosis, and cancer, cause changes in tissue stiffness, resulting in a smaller strain in hard tissues than in soft tissues [21,22,23]. It is possible to evaluate strain elastography analysis quantitatively by looking at the strain ratio and strain histogram, or qualitatively based on the color map distribution.
Qualitative Strain Elastography

In qualitative elastography, the structures in the B-mode image are quantified for their compression-induced deformation, with the degree of deformation serving as a measure of tissue stiffness [21]. When conducting the evaluation, the probe must be fastened to the wall with just the right amount of pressure to produce an ideal and stable B-mode image. The targeted lesion and any nearby tissues are included in the region of interest (ROI) that is manually chosen for the elastographic analysis.

For elastographic registration, the highest sensitivity is advised. A color map (red-green-blue) is used to represent elasticity (on a scale of 1–255), with hard tissue being represented by dark blue, medium hard by cyan, intermediately hard by green, medium soft by yellow, and soft by red. By overlaying the color pattern on a typical B-mode image, the elastography pattern is made clear. The standard presentation format is a two-panel image with the elastographic image on the left and the traditional grey-scale B-mode image on the right (Figure 1). By using frame average analysis in elastographic software, bias in manual image selection is avoided. Additionally, the system picks the best frames for analysis [21,23]. Table 1 defines the elastographic patterns and their meaning.
Quantitative Strain Elastography

There are two options for quantitative elastography evaluation: a strain histogram and strain ratio. In both cases, the first step is to obtain stable elastographic images, as previously described.
-*Strain Histogram*

The strain histogram is a graphical representation of the color distribution in a selected image field. The strain histogram is based on the qualitative elastographic data for a manually selected ROI within the standard elastography image. The x-axis represents the elasticity of the tissue, from 0 (hardest) to 255 (softest). The y-axis represents the number of pixels in each elasticity level in the ROI. The mean value of the histogram corresponds to the global hardness or elasticity of the lesion [21] (Figure 2B). Table 1 summarizes the strain histogram values and their correlation.
-*Strain ratio*

Qualitative elastographic patterns are relative to some extent. The calculation of strain ratio, which analyzes the elastographic picture of the target lesion in relation to the surrounding tissues, is an attempt to address this problem. As for the strain histogram, strain ratio calculation is based on standard qualitative elastographic data. Two different areas (A and B) are selected for quantitative elastographic analysis. Area A is selected to include as much of the target lesion as possible without including the surrounding tissues. Area B is selected within a soft (red) reference area outside the target lesion, preferably the gut wall. The strain ratio is calculated as the quotient of B/A (Figure 2A). An assumption of this method is that the investigated disease does not significantly alter the hardness of the reference connective or fat tissues [23]. Table 1 summarizes the strain ratio values and their correlation.

### 2.2. Shear Wave Elastography

Since it was introduced in 2019, EUS-guided shear wave elastography (SWE) is a non-invasive medical imaging technique used to evaluate tissue stiffness, using absolute values to determine the tissue’s elasticity [24]. It works by measuring the speed of shear waves generated by acoustic radiation force within the tissue. The stiffer the tissue, the faster the shear waves propagate.

This modality uses a doppler-like ultrasound technique to track the spread of shear waves and gauge their velocity. Theoretically, shear waves propagate more quickly in tissues with higher tissue elasticity. The shear-wave velocity (Vs), a measure of an elastic module, is taken in a target lesion. To determine whether shear-wave propagation is correctly detected and whether unnecessary components other than those generated by shear-wave propagation existed in the ROI by predefined rejection conditions, the reliability index is used to calculate the percentage of the net amount of effective shear-wave velocity (VsN, percent). The ROI measures 5 mm in height by 10 mm in width, and it is placed close to the tissue or lesion being examined while avoiding as many calcifications, blood vessels, and cystic components as possible (Figure 3). To prevent breathing artifacts, the measurement is conducted when there is the least amount of respiratory fluctuation [24,25,26].

## 3. Clinical Applications of EUS Elastography

The two primary indications for EUS elastography are the evaluation of solid pancreatic lesions and enlarged lymph nodes; however, over the years, more indications have been established for EUS-guided elastography. 

## 4. Pancreatic Diseases

EUS is considered a reference method for the diagnosis and staging of inflammatory, cystic, and neoplastic lesions of the pancreas [15,27,28,29]. Nevertheless, the capability of conventional B-mode EUS to differentiate between benign and malignant pancreatic lesions can be far from optimal in certain clinical situations. For example, its overall accuracy in differentiating between pancreatic cancer and focal pancreatitis in advanced CP is not higher than 75% [15]. EUS elastography may be helpful in these situations.

### 4.1. Differential Diagnosis of Solid Pancreatic Lesions

EUS-guided elastography is highly accurate for the differential diagnosis of solid pancreatic tumors. The first study of EUS elastography in pancreatic solid lesions was published by Giovannini et al. [30]. Specific elastographic patterns have been described, following this first publication. Normal pancreatic parenchyma typically has a homogeneous green pattern; inflammatory pancreatic masses typically have a heterogeneous, predominantly green pattern with faint yellow and red lines; pancreatic malignant tumors typically have a heterogeneous, predominantly blue pattern with small green areas and red lines and a geographic appearance and pancreatic neuroendocrine malignant lesions typically have a homogeneous blue pattern (Table 1). Thus, compared to the surrounding pancreatic parenchyma, pancreatic cancer (PC) exhibits a pattern that is almost unmistakably very stiff and blue in color (Figure 4). In most instances, the elastography appearance of chronic pancreatitis (CP) differs from that of PC [23]. It is crucial to draw attention to the distinctive pattern in autoimmune pancreatitis cases because this condition exhibits a recognizable diffuse stiff pattern throughout the entire pancreatic parenchyma, not just at the mass [31,32]. We have published our own experience with qualitative EUS elastography in 130 patients with solid pancreatic masses and 20 controls, and using this classification, the sensitivity, specificity, positive and negative predictive values, and overall accuracy of EUS elastography for detecting malignancy were 100, 85.5, 90.7, 100, and 94.0%, respectively [33]. However, not all studies have shown the same accuracy, with some of them leading to low specificity, and problems to perform an adequate elastographic evaluation. These problems are primarily related to large lesions (>35 mm), in which it may be difficult to include the entire lesion and enough surrounding tissue in the analyzed ROI, lesions that are far from the transducer, and lesions that contain fluid (vessels, cysts, ducts, etc.) in the ROI. Importantly, the latest advances in elastography software are helping to overcome these problems.

Another important fact, coming from some key studies is related to the very good interobserver agreement for determining the malignant potential of lesions evaluated, with kappa scores around 0.7 [33,34,35]. 

In terms of quantitative elastography, compared to inflammatory masses and normal parenchyma, malignant pancreatic tumors, and neuroendocrine lesions produce higher strain ratios and lower strain histograms. It has been recommended that a mean strain histogram value of or a strain ratio >10 is used. <50 is associated with malignancy, whereas the presence of a strain ratio < 10 or a mean strain histogram value of >50 is linked to benign diseases (Table 1) (Figure 5). We have published the strain-ratio results of 86 consecutive patients with pancreatic solid lesions (49 adenocarcinomas, 27 inflammatory masses, 6 malignant neuroendocrine tumors, 2 metastatic oat cell lung cancers, 1 pancreatic lymphoma, and 1 pancreatic solid pseudopapillary tumor) and 20 controls. The strain ratio was significantly higher among patients with malignant pancreatic tumors than in those with inflammatory masses. Normal pancreatic tissue showed a mean strain ratio of 1.68 (95%CI 1.59–1.78). Inflammatory masses presented a strain ratio (mean 3.28; 95%CI 2.61–3.96) significantly higher than that of the normal pancreas (*p* < 0.001), but lower than that of pancreatic adenocarcinoma (mean 18.12; 95%CI 16.03–20.21) (*p* < 0.001). The highest strain ratio was found among endocrine tumors (mean 52.34; 95%CI 33.96–70.71). The sensitivity and specificity of the strain ratio for detecting pancreatic malignancies using a cut-off value of 6.04 were 100 and 92.9%, respectively, exceeding the accuracy obtained with qualitative elastography [36]. More recently we have published our experience with both strain ratio and strain histogram in solid pancreatic lesions. A positive elastography finding (defined as strain ratio >10 or strain histogram level < 50) classified lesions as malignant with 100% sensitivity and 92.3% specificity. In this study, no differences were observed by using either strain ratio or strain histogram [37].

According to a recent multicenter trial, 50% of solid pancreatic lesions under 15 mm were found to be soft, and the likelihood that a soft lesion would become malignant was very low. Therefore, due to its extremely high negative predictive value for malignancy, EUS-guided elastography may be particularly useful for small pancreatic tumors [38]. Elastography’s ability to locate blue spots (hard tissue) inside the mass-forming CP and guide the sampling area is another benefit. The impact of EUS-guided sampling guided by elastography, however, has not been demonstrated.

The effectiveness of EUS-guided elastography as a diagnostic tool for identifying malignant pancreatic tumors has been assessed by various meta-analyses. In this clinical application, elastography displayed high sensitivity (92–98%) but low specificity (67–76%) [39,40,41,42].

SWE has been examined for solid pancreatic tumors in recent studies. SWE and SH were compared by Ohno et al. The vs. (m/s) values for PC, pancreatic neuroendocrine tumors, mass-forming pancreatitis, and metastatic tumors were 2.19, 1.31, 2.56, and 1.58, respectively. Based on the disease, vs. did not reveal any appreciable differences. The average strain values were 45.5 for PC, 47.3 for neuroendocrine tumors, and 74.5 for the inflammatory process. However, the mean strain value was significantly lower in PC cases (45.4 vs. 74.5: *p* = 0.0007). Tissue elasticity between PC and inflammatory lesions was compared vs. did not show a significant difference (*p* = 0.5687) [43]. Figure 6 shows an example of SWE from pancreatic cancer.

We summarize in Figure 1 the diagnostic accuracy of EUS-guided elastography in solid pancreatic tumors [30,34,36,38,43,44,45,46,47,48,49,50,51,52,53,54].

The staging of Yamada et al. also demonstrated a potential role of elastography in the staging of PC, mainly for vascular compromise. On 44 multidetector computer tomography (MDCT) yielded a sensitivity, specificity, and accuracy of 0.733, 0.697, and 0.708; in EUS B-mode, they were 0.733, 0.606, and 0.646. These results improved to 0.900, and 0.906 after an elastographic analysis was conducted. Sensitivity, specificity, and accuracy were 0.556, 0.750, and 0.690 on MDCT in 27 subjects with a tumor contacting a vessel without vascular obstruction or stenosis; 0.667, 0.700, and 0.690 in EUS B-mode and 0.889, 0.850, and 0.862 in elastography [55]. 

### 4.2. Chronic Pancreatitis

According to our observations, CP typically exhibits a heterogeneous green predominant pattern, as opposed to the homogeneous green pattern displayed by a normal pancreas. Comparing a normal pancreas to inflammatory and malignant lesions, a normal pancreas typically exhibits lower strain ratio levels. We discovered a significant direct linear correlation (r = 0.813; *p* < 0.0001) between the quantity of EUS criteria for CP and the strain ratio as a gauge of the pancreatic fibrosis severity in CP. The accuracy of EUS elastography for the diagnosis of CP was 91.1 percent (cut-off strain ratio of 2.25), and the area under the ROC curve was 0.949 (95%CI 0.906–0.982). The strain ratio varied significantly across the various Rosemont classification categories (1.80 normal pancreas, 2.40 indeterminate group, 2.85 suggestive of CP, 3.62 consistent with CP, *p* < 0.001) (Figure 7) [56]. Itoh et al. demonstrated a very strong correlation between the histological finding, measures by the standardized fibrosis score, and the elastographic evaluation. The areas under the ROC curves for the diagnosis of mild or higher-grade fibrosis, marked or higher-grade fibrosis, and severe fibrosis were 0.90, 0.90, and 0.90, respectively [57]. Elastography is also useful for establishing the severity of the disease, so the higher the strain ratio, the higher the possibility of exocrine pancreatic insufficiency [58]. Our group has developed the EUS multimodal test for the evaluation of suspected CP. This method includes EUS criteria for CP, strain ratio, and endoscopic pancreatic function test (ePFT) with the distensibility of the main pancreatic duct. The strain ratio was abnormally high in all patients. Peak bicarbonate concentration was decreased by 81.1% and compliance was reduced by 77.3%. The presence of abnormal morphological and functional evaluation of the pancreas could support the clinical suspicion of early CP in the appropriate clinical setting [59]. We have recently published that the degree of pancreatic fibrosis as evaluated by elastography correlates with the ePFT in patients with clinical suspicion of CP and inconclusive EUS findings (r = 0.715, *p* < 0.0001). Using the ePFT as gold-standard, EUS-guided elastography yielded a diagnostic accuracy of 93.4% for CP [60].

SWE has also been tested in CP. Yamashita et al. showed a correlation between the shear-wave velocity and the Rosemont classification and certain EUS features of CP. Shear-wave velocity was consistent with CP (2.98 m/s) and was suggestive of CP (2.95 m/s). The results were significantly higher than those found for normal tissue (1.52 m/s). This methodology also showed high accuracy for diagnosing CP, with the area under the curve of 0.97. The velocity cut-off of 2.19 m/s showed 100% sensitivity and 94% specificity for CP [61]. More recently, another study from Yamashita et al. showed that shear wave elastography values positively correlated with the severity grades of Rosemont classification and Japan Pancreatic Society criteria, but strain elastography values did not [62]. 

Age-related changes (similar in B-mode to those described for CP) and real CP can also be distinguished using elastography. Janssen et al. have written about their experiences with 26 patients with diffuse CP (group 3), 46 individuals with healthy pancreas (grouped into individuals up to 60 years of age (group 1) and those over 60 (group 2)), and both. In groups 1–3, the pancreatic body’s mean strain values were 110.2, 80.0, and 32.4, respectively. The area under the ROC curve for differentiating between CP and a healthy pancreas in individuals over the age of 60 was 0.993 at a cut-off value of 50. According to these findings, quantitative elastography demonstrates that the pancreas continues to be softer than in CP even though it becomes noticeably harder with age [63].

## 5. Lymph Nodes

Several studies have shown the usefulness of EUS-guided elastography in the evaluation of lymph nodes (LN), reviewed by Dietrich et al. [64]. 

Giovannini et al. carried out the initial studies. The sensitivity and specificity for malignancy in their initial study were 100% and 50%, respectively [30]. A subsequent multicenter trial revealed sensitivity, specificity, and overall accuracy for malignancy of 91.8%, 82.5%, and 88.1%, respectively when benign lesions tests were viewed as negative and indeterminate and malignant lesions tests as positive. The interobserver agreement yielded a κ score of 0.657 for the detection of malignant LN [34]. 

According to the research by Janssen et al. diagnostic yield ranged from 81.8 to 87.9% for benign LN and from 84.6 to 86.4% for malignant LN. Interobserver agreement yielded a κ = 0.84 [65]. At our institution, a study was conducted on qualitative EUS elastography for the assessment of LN. On final diagnosis, 63 LN from 57 patients were included, of which 54 mediastinal and 9 abdominal LN were malignant and 31 benign. A primarily blue pattern, a pattern that is primarily green, and a pattern that is mixed (blue and green, without predominance) were all recognized as distinct elastographic patterns; 7 of the 31 malignant LN displayed a mixed pattern, while 24 displayed a mostly blue pattern. With a green pattern, no malignant LN were seen. The predominant elastographic patterns of 23 of the 32 benign LN were green, 2 were blue, and 7 were mixed. Thus, a benign histology in LN that exhibits a green pattern on elastography is 100% likely to be benign (Figure 8), whereas a malignant histology with a predominantly blue pattern is 92.3% likely to be malignant. The likelihood of malignant histology was 50% in cases with mixed patterns on elastography.

More studies have been performed using quantitative elastography (Figure 9). Saftoiu et al. [66] with the use of hue histogram, yielded a sensitivity, specificity, and accuracy in determining malignant potential were 85.4, 91.9, and 88.5%, respectively, using a cut-off point of 166. 

Paterson et al. [67] presented findings showing the effectiveness of this technique in the assessment of patients with LN when researching patients with esophageal and gastric cancer. In 53 LN, they carried out quantitative EUS elastography and EUS-guided sampling. The area under the ROC for the strain ratio was 0.87 (*p* < 0.0001). Sensitivity, specificity, positive predictive value, and negative predictive value was 83, 96, 95, and 86%, respectively, for determining the malignant potential of LN. The overall accuracy of the strain ratio was 90%. Puga-Tejada et al. analyzed 121 patients, founding with the strain ratio analysis sensitivity and specificity for malignancy of 90.9% and 95.2%, respectively [44]. Not all studies, however, had positive findings. Larsen et al. reported their observations regarding the value of strain ratio analysis of LN in patients with upper gastrointestinal cancer. In contrast to the various elastographic modalities, the sensitivity of EUS for differentiating between malignant and benign LN was 86%. The specificity of EUS was only 71%, which was less than that of EUS-guided elastography, which was between 82 and 85% [68]. 

We resume in Figure 2 the studies analyzing the diagnostic yield of EUS-guided elastography in determining the malignancy of LN [30,34,44,65,66,67,68,69,70].

One meta-analysis with seven studies, 368 patients, and 431 LN in total, looked into the distinction between benign and malignant nodes. The combined EUS elastography sensitivity and specificity for differentiating between benign and malignant LN were 88 and 85%, respectively, 0.9456 was the area under the ROC curve. The authors concluded that EUS-guided elastography is a promising, noninvasive technique for the differential diagnosis of malignant LN, and it may prove to be a useful adjunct technique to EUS-guided sampling [71].

## 6. Gastrointestinal Lesions

Differentiating gastrointestinal stromal tumors (GIST) from other mesenchymal lesions such as leiomyoma or schwannoma when dealing with subepithelial lesions is crucial. Although EUS-guided sampling has demonstrated good accuracy, accessing small lesions is very challenging [72]. For the management of these lesions, imaging-based differentiation is, therefore, useful.

Tsuji et al. evaluated 25 gastric subepithelial lesions for providing the elastographic score. According to their research, GIST is portrayed as “hard” tissues when compared to other subepithelial lesions [73] (Figure 10). On the other hand, Ignee et al. by using a pattern diagnosis and an elastic score, found it difficult to distinguish GIST from benign leiomyoma [74]. It is still worth looking into whether EUS-guided elastography will be useful in this field in the future.

## 7. Transrectal EUS Elastography

For the diagnosis and assessment of prostate cancer, rectal cancer, inflammatory bowel disease, and fecal incontinence, transrectal EUS elastography has been studied. Elastography is superior to transrectal EUS alone in the diagnosis of prostate cancer, and it increases the specificity of prostate biopsies by highlighting regions with a high degree of malignancy suspicion. In patients with a clinical suspicion of prostate cancer, the specificity and sensitivity of transrectal elastography ranges from 62 to 87% and 68 to 92%, respectively, in the diagnosis of prostate cancer [75,76,77,78].

One study involving 69 patients with rectal tumors evaluated the use of transrectal elastography to distinguish between benign and malignant rectal tumors. With a sensitivity of 93%, specificity of 96%, and accuracy of 94%, quantitative elastography using the strain ratio distinguished between adenomas and adenocarcinomas [79,80]. The strain ratio of EUS evaluation of rectal wall thickness was investigated for the diagnosis of inflammatory bowel disease and the distinction between Crohn’s disease and ulcerative colitis. Compared to controls and ulcerative colitis patients, patients with Crohn’s disease had noticeably higher strain ratios [81]. Allgayer et al. analyzed the elastography of anal sphincters in 50 patients with fecal incontinence, and no association between the patients’ functional and clinical parameters and the sphincter’s elastographic appearance was discovered [82].

## 8. Other Indications

Considering present indications for EUS, EUS-guided elastography may be useful for the study of solid lesions in adrenal glands, by distinguishing adenomas from metastases. Our preliminary unpublished data support this idea. Another potential indication for elastography is the evaluation of liver fibrosis [83] and in the differentiation between benign and malignant solid liver lesions [84]. 

## 9. Conclusions

EUS-guided elastography is considered today an excellent tool, capable of differentiating fibrotic/inflammatory tissues from malignant lesions. It has been able to demonstrate that it can distinguish benign and malignant solid pancreatic tumors and LN with high accuracy, as well as to differentiate normal pancreas from early CP. Future research will keep updating and defining the role of EUS elastography in clinical practice.

## Data Availability

Not applicable.

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
