# Peer review of "Endoscopic Ultrasound (EUS) Guided Elastography"

_diagnostics, 2023, doi:10.3390/diagnostics13101686_

Round 1
Reviewer 1 Report
The authors reviewed EUS guided elastography. The manuscript is well written and provides a comprehensive overview of the use of EUS-guided elastography for pancreatic diseases, lymph nodes and gastrointestinal lesions. However, there are so many mismatches in the main text and Graphics/Figures.
1. Line 91: In the main text, the authors described that Graphic 1 define the elastographic patterns and its meaning. But Graphic 1 reveals studies analyzing the accuracy of EUS-guided elastography in solid pancreatic lesions. Please correct.
2. Line 109, 121: The authors described that Graphic 1 summarizes the strain histogram values and its correlation. But I could not find the relevant graphics. Please correct.
3. Figure 2 has two pictures. Please insert A and B in the pictures.
4. Line 184: The authors mentioned graphic 1 that may be related to the color pattern. I cannot find the relevant graphics.
5. Line 212: The authors mentioned graphic 1 again. I cannot fine the relevant graphics.
6. Line 253: The authors described that we summarize in Graphic 2 the diagnostic accuracy of EUS guided elastography in solid pancreatic tumors. But Graphic 2 reveals studies analyzing the accuracy of EUS-guided elastography in the differential diagnosis of lymph nodes. Please correct.
7. Line 267: Please write the full name of MDCT first.
8. Line 343: In the main text, figure 8 seems to show the elastography for a lymph node (LN). But figure 8 shows elastography for chronic pancreatitis.
9. Line 399: In the main text, figure 9 seems to show the elastography for GIST. But figure 9 shows elastography for a LN
10. Line 431: Considering present indications for EUS, EUS-guided elastography may be useful for the study of solid lesions at suprarenal glands, by distinguishing adenomas from metastases. Please show the reference for above sentence.
Author Response
The authors reviewed EUS guided elastography. The manuscript is well written and provides a comprehensive overview of the use of EUS-guided elastography for pancreatic diseases, lymph nodes and gastrointestinal lesions. However, there are so many mismatches in the main text and Graphics/Figures.
- Line 91: In the main text, the authors described that Graphic 1 define the elastographic patterns and its meaning. But Graphic 1 reveals studies analyzing the accuracy of EUS-guided elastography in solid pancreatic lesions. Please correct.
Thanks for the comment, we have made the correction in the manuscript.
- Line 109, 121: The authors described that Graphic 1 summarizes the strain histogram values and its correlation. But I could not find the relevant graphics. Please correct.
Thanks for the comment, we have made the correction in the manuscript.
- Figure 2 has two pictures. Please insert A and B in the pictures.
We have modified the Figure according to the recommendation.
- Line 184: The authors mentioned graphic 1 that may be related to the color pattern. I cannot find the relevant graphics.
Thanks for the comment, we have made the correction in the manuscript.
- Line 212: The authors mentioned graphic 1 again. I cannot fine the relevant graphics.
Thanks for the comment, we have made the correction in the manuscript.
- Line 253: The authors described that we summarize in Graphic 2 the diagnostic accuracy of EUS guided elastography in solid pancreatic tumors. But Graphic 2 reveals studies analyzing the accuracy of EUS-guided elastography in the differential diagnosis of lymph nodes. Please correct.
Thanks for the comment, we have made the correction in the manuscript.
- Line 267: Please write the full name of MDCT first.
Thanks for the comment, we have made the correction in the manuscript.
- Line 343: In the main text, figure 8 seems to show the elastography for a lymph node (LN). But figure 8 shows elastography for chronic pancreatitis.
Thanks for the comment, we have made the correction in the manuscript.
- Line 399: In the main text, figure 9 seems to show the elastography for GIST. But figure 9 shows elastography for a LN
Thanks for the comment, we have made the correction in the manuscript.
- Line 431: Considering present indications for EUS, EUS-guided elastography may be useful for the study of solid lesions at suprarenal glands, by distinguishing adenomas from metastases. Please show the reference for above sentence.
As stated in the manuscript, this is based on our personal (long) experience, haven’t published the results yet.
Reviewer 2 Report
General comments:
Interesting, timely topic with well rounded review of EUS guided elastography.
Grammar revisions required.
Specific comments:
I don't feel that the difference between shear wave and strain elastography has been described clearly and requires some revision.
Figure 9 (page 13) : Has been described as a predominantly green pattern. The images shows a predominantly blue pattern.
Page 14 line 385: please change the word "ones" to "nodes."
Page 14 line 393: do not capitalize the word "schwannoma"
Page 15 line 432: please change "suprarenal glands" to "adrenal glands."
Conclusion (page 16): Grammar revision required.
References: adequate
Figures: The quality of the images is very good.
Author Response
Interesting, timely topic with well rounded review of EUS guided elastography.
Grammar revisions required.
Thanks for the comment, we have revised the grammar and English as requested
Specific comments:
I don't feel that the difference between shear wave and strain elastography has been described clearly and requires some revision.
We have included an additional description of SWE, with the aim to clarify the difference between both methods.
Figure 9 (page 13) : Has been described as a predominantly green pattern. The images shows a predominantly blue pattern.
Thanks for the comment, we have made the correction in the manuscript.
Page 14 line 385: please change the word "ones" to "nodes."
Thanks for the comment, we have made the correction in the manuscript.
Page 14 line 393: do not capitalize the word "schwannoma"
Thanks for the comment, we have made the correction in the manuscript.
Page 15 line 432: please change "suprarenal glands" to "adrenal glands."
Thanks for the comment, we have made the correction in the manuscript.
Conclusion (page 16): Grammar revision required.
We have revised the grammar and English as requested
References: adequate
Figures: The quality of the images is very good.
Round 2
Reviewer 1 Report
The paper was revised properly following the reviwer's comment.
Author Response
Thanks for the comments
Reviewer 2 Report
The authors have done a great job with all of the changes that I have recommended, except for editing grammar, especially in the introduction. I would recommend running the manuscript through the program Grammarly. The grammar edits can then be made quickly, effectively, and free of charge. These minor grammar issues affect the readability of the manuscript and take away from the otherwise good job of the authors.
Two examples from the introduction page 1:
The second sentence, line 10 "Over the years, new technologies have been developed because to improve and overcome..." Please rephrase: "Over the years, new technologies have been developed to improve and overcome..."
Line 19 "...there are well establish indications..." Please change to "...there are well established indications..."
Author Response
The authors have done a great job with all of the changes that I have recommended, except for editing grammar, especially in the introduction.
Thanks again for the comment.
In order to improve grammar, we have followed your recommendation, and really believe we have improved accordingly.
Two examples from the introduction on page 1:
In the second sentence, line 10 "Over the years, new technologies have been developed because to improve and overcome..." Please rephrase: "Over the years, new technologies have been developed to improve and overcome..."
Line 19 "...there are well establish indications..." Please change to "...there are well established indications..."
According to the recommendations, we have made the proper changes